# Numerical Simulation of an Online Cotton Lint Sampling Device Using Coupled CFD–DEM Analysis

**Peiyu Wang** [1,2,3], **Huting Wang** [1,2,3,*], **Ruoyu Zhang** [1,2,3], **Rong Hu** [1,2,3], **Beibei Hao** [1,2,3] **and Jie Huang** [1,2,3]

1   College of Mechanical and Electrical Engineering, Shihezi University, Shihezi 832003, China; 20212009003@stu.shzu.edu.cn (P.W.); hurong@shzu.edu.cn (R.H.)
2   Key Laboratory of Northwest Agricultural Equipment, Ministry of Agriculture and Rural Affairs, Shihezi 832003, China
3   Key Laboratory of Modern Agricultural Machinery Corps, Shihezi 832003, China
*   Correspondence: xgb@shzu.edu.cn

**Abstract:** Cotton processing is the process of converting harvested seed cotton into lint by cleaning, ginning, and cleaning the lint. The real-time acquisition of lint parameters during processing is critical in improving cotton processing quality and efficiency. The existing online inspection system cannot realize quantitative sampling detection, resulting in large fluctuations in the detection of moisture rate, and the impurity content of lint can only be measured according to the number of impurity grains and the percentage of impurity areas. This research developed a quantitative sampling device for cotton lint processing that can collect the right number of cotton samples and obtain the weight of the samples, laying the foundation for the accurate detection of cotton lint dampness and impurity rates. This research aimed to develop an online quantitative sampling device with a sampling plate as its core. The quantitative sampling procedure, consisting of a gas–solid two-phase flow in a cotton pipeline, was numerically simulated and experimentally analyzed using computational fluid dynamics (CFD) and the discrete element method (DEM). According to the coupling results, the maximum pressure differential between the top and bottom regions of the sampling plate when conveying was 1024.45 Pa. This pressure is adequate to allow for cotton samples to accumulate on the sampling plate. Simultaneously, the steady conveying speed of lint is 59.31% of the unloaded conveying wind speed, providing a theoretical foundation for the sampling time of the quantitative sample device in the processing chain. The results from testing the prototype indicate that the quantitative sampling device in the cotton flow can effectively perform the quantitative sampling of cotton lint under uniform conditions, with a sampling pass rate of 84%.

**Keywords:** cotton pipeline; sampling plate; negative-pressure transportation; computational fluid dynamics; discrete element method

## 1. Introduction

Online lint quality inspection in the processing chain is a critical aspect influencing cotton processing quality. According to National Bureau of Statistics data, the national cotton planting area in 2022 was 3000.3 thousand hectares, with a production of 5.977 million tons. Of these, 5.391 million tons were produced in Xinjiang, China's largest cotton production area, accounting for 90.2% of the country's cotton production. With the popularization of cotton production mechanization in Xinjiang, machine-picked cotton also introduces certain quality problems [1]; the processing of cotton lint and online detection of dampness and impurity rate can provide effective feedback information for the adjustment of the process parameters of seed cotton, which is very important for the improvement of quality and efficiency in the cotton industry.

Cotton moisture and impurity rate are major factors influencing the stability and quality of the cotton processing system. Domestic and foreign scholars have conducted

extensive research on cotton moisture and impurity rate detection. Regarding moisture detection, foreign Uster cotton online quality testing equipment was installed in the lint pipeline on an online sampling workstation to obtain cotton water content, impurities, and color information [2,3]. The Zhengzhou Cotton and Jute Engineering Technology and Design Research Institute has developed an online determination device for lint moisture regain. This device uses the resistance method to detect the moisture rate of cotton in real time. It consists of a detection electrode set, deflector plate, microprocessor, and display module. This device helps to improve the stability of cotton pressure and compactness during the detection process, resulting in more-accurate moisture rate measurements [4,5]. Xiao Zhonggao et al. [6] revealed a new device designed for the online testing of cotton. This device consists of a porous sampling plate, a sampling rotating shaft, and a motor. The sampling plate is positioned perpendicular to the direction of the cotton flow and driven by the motor to perform automatic sampling. This device enables automatic sampling and pressure during the online testing process, providing technical support for measuring the dampness rate of cotton. In terms of impurity detection, Wan Long et al. [7] created a machine-picked seed cotton acquisition link impurity detection system that used RGB double-sided imaging to obtain a single image then analyzed the image impurities to determine the proportion of the predicted impurity rate of seed cotton samples. Tian Hao [8] employed image threshold segmentation and connection region analysis techniques to quantify the pixel area occupied by cotton and impurities. Additionally, regression analysis was utilized to forecast the impurity rate of cotton. In their study, Wu Tingrong et al. [9] employed two distinct techniques to perform image segmentation, namely the maximum interclass variance approach and edge detection in conjunction with multiple operators. They then determined the impurity rate by calculating the pixel area share of the impurity region. In short, the study and implementation of technology for detecting moisture content and trash content in lint are quite advanced. However, due to the complex flow field environment in pipelines, the existing online cotton lint inspection device is unable to realize the quantitative removal of cotton samples and is unable to obtain the weight of cotton samples in the online inspection of cotton lint moisture and impurity content during processing. The result is that quality indicators such as cotton lint moisture and impurity cannot be quantitatively detected. The issue of testing accuracy being significantly influenced by variations in sampling weight requires immediate attention and resolution.

The utilization of computational fluid dynamics (CFD) has become a prominent approach in the examination of fluid motion. Similarly, the discrete element method (DEM) has gained significant popularity in the analysis of particle collision. By combining CFD and the DEM, it becomes possible to accurately monitor the movement of individual particles within a gas–solid flow system, thereby acquiring a substantial quantity of microscopic data [10–13]. The primary emphasis in the numerical simulation of the gas–solid two-phase flow of agricultural materials is on near-spherical particles, including wheat, soybeans, corn, and similar materials. The materials are typically transformed into spherical particles through the utilization of equivalent diameter, material characteristic parameters, contact parameters, and other relevant factors [14–19]. Subsequently, numerical simulations of gas–solid two-phase flow are conducted using computational fluid dynamics (CFD) software and the discrete element method. This approach has yielded significant advancements in the field of agricultural equipment development [20–22]. Huang Zhenyu et al. [23] analyzed the gas–solid two-phase flow of vegetable seeds in the separation chamber of a wind sifter using the RNG k-ε turbulence model and the DMP discrete-phase model of FLUENT 2021 R1 software. Zhang Xian et al. [24] simplified gross tea material into spherical particles and used hydrodynamic methods to study the internal flow field and material motion trajectory of a tea wind separator. Liu Jia et al. [25] simulated the working process of a mechanical–pneumatic combined precision seeder based on a coupled CFD–DEM method, and the model was built with non-spherical virtual corn grains by using bonding and API replacement in EDEM 2022 software. Cotton is classified as a flexible flocculent material, characterized by intricate mechanical properties and nonlinearity in deformation. Conse-

quently, investigating the motion characteristics of cotton through numerical calculations has posed a significant challenge in the field of computer numerical simulation.

This paper presents a novel approach utilizing the CFD–DEM coupled simulation method to develop a quantitative sampling detection device for the processing of cotton lint. The proposed method involves conducting numerical simulations to analyze the movement of lint particles in the flow field during pneumatic conveying and to assess the particle stacking state within the sampling device. The outcomes of this study aim to offer a technical foundation for the advancement and optimization of a quantitative detection system, specifically for evaluating the dampness and impurity content of cotton lint during processing.

## 2. Machine Structure and Working Principle

### 2.1. Overall Structure

The lint quantitative sampling device as part of the processing chain mainly consists of sampling and weighing components. The sampling components mainly include a servo motor, reducer, coupling, rotary shaft, sampling plate, and air nozzle. The weighing component mainly comprises a cotton collection box and a load cell. The specific structure is shown in Figure 1.

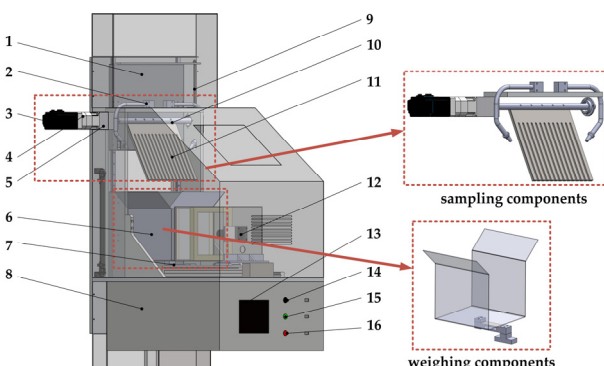

**Figure 1.** Lint quantitative sampling device structure diagram as part of the processing chain: (1) negative-pressure adjustment plate, (2) nozzle, (3) servo motor, (4) moderator, (5) coupling, (6) cotton collection box, (7) weighing sensors, (8) electric cabinet, (9) cylinder, (10) rotary axis, (11) sample plate, (12) information detection device, (13) touchscreen, (14) power switch, (15) start button, (16) emergency switch.

The servo motor is connected to a gearbox (ratio = 1:25) for torque transmission; the rotary axis is connected to the moderator using a coupling to realize the rotating action; the sample plate is solidly connected to the concave surface of the rotary axis and rotates with the rotary axis to complete the sampling action; the negative-pressure adjustment plate moves up and down, driven by a pneumatic cylinder to realize a negative-pressure barrier effect and provide a stable environment for cotton sample weighing; the cotton collection box is connected to weighing sensors to obtain the weight of the sample; and the weighing sensors are mounted on the base plate to ensure weighing stability.

### 2.2. Principle of Operation

A quantitative sampling device was installed in a cotton processing plant's cotton lint pipeline with the sampling components to complete the sampling action of the processing of lint through the detection of lint's dampness and impurity rate depending on the size of cotton samples' weight to provide the basis for accurate detection. The working principle is shown in Figure 2.

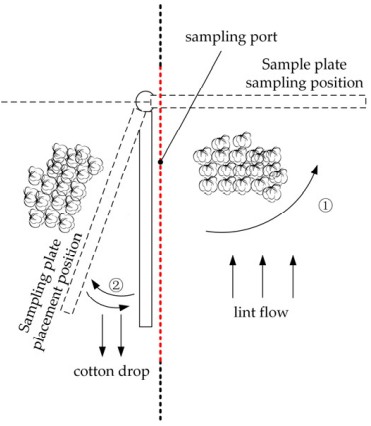

**Figure 2.** Sampling schematic.

To start sampling, the negative-pressure adjustment plate moves up, the sampling port opens, the servo motor rotates counterclockwise by 90°, the sampling plate is driven by the rotary axis to the vertical position with the side wall of the cotton pipeline (hanging inside the pipeline), and the cotton starts to pile up on the sampling plate. At the end of sampling, the servo motor rotates 120° clockwise, the sample plate is driven by the rotary axis to the quantitative sampling device inside, and the pile of cotton samples on the sampling plate drops into the device inside; more specifically, collecting cotton samples, the negative-pressure adjustment plate moves down, the sampling port is closed, the airflow between the sampling plate and the cotton transport pipeline is blocked, and the cotton samples piled up on the sampling plate fall into the cotton collection box. As for cotton sample weighing, the cotton falls into the collection box through the weighing sensor to obtain the actual weight of samples, to complete the quantitative sampling work. Eventually, the information detection device moves forward and compacts the cotton sample, completing the accurate detection of the sample's moisture regain and trash content based on a known weight.

### 3. CFD–DEM Numerical Simulation of Quantitative Sampling Device

*3.1. Mathematical Model*

3.1.1. Gas-Phase Control Equations

Cotton pneumatic conveying link airflow is transported from the cotton collection inlet to the return duct, which is considered to be a continuous gas phase following the mass conservation law and momentum conservation law [11,26]. The governing equations for the conservation of mass and momentum of an incompressible viscous fluid can be expressed as follows:

$$\frac{\partial(\varepsilon_g \rho_g)}{\partial t} + \nabla(\varepsilon_g \rho_g v_g) = 0 \tag{1}$$

$$\frac{\partial(\varepsilon_g \rho_g v_g)}{\partial t} + \nabla(\varepsilon_g \rho_g v_g \otimes v_g) = -\varepsilon_g \nabla P + \varepsilon_g \nabla \tau + \varepsilon_g \rho_g v_g - R_{gp} \tag{2}$$

included among these:

$$R_{gp} = \sum_{i=1}^{n} F_{pi} / \Delta V \tag{3}$$

where $\rho_g$ is the density of air, $v_g$ is the air velocity, $\varepsilon_g$ is the gas volume fraction, $P$ is the gas pressure, $\tau$ is the viscous stress tensor, $g$ is the gravitational acceleration, $R_{gp}$ is the momentum exchange between solid and gas phases of the unit grid, $F_{pi}$ is the combined force acting on lint particles, $n$ is the number of particles in a given mesh, $\Delta V$ is the volume of units, and $t$ is the timing.

3.1.2. Solid-Phase Governing Equations

The investigated particle motion is solved by the discrete element method (DEM), where the cotton cleaned by the lint cleaner is subjected to several forces such as traction, gravity, buoyancy, Saffman force (shear lift), and Magnus force (rotational lift) in the pneumatic conveying link. The lint particles' motion is described by Newton's second law [11] as:

$$m_p \frac{dv_p}{dt} = F_D + F_{GB} + F_{Sa} + F_{Ma} \tag{4}$$

$$I_p \frac{d\omega_p}{dt} = T \tag{5}$$

where $m_p$ is the lint particle quality, $F_D$ is the motive force, $F_{GB}$ is the combined force of gravity and buoyancy, $F_{Sa}$ is the shear lift, $F_{Ma}$ is the rotational lift, $I_p$ is the moment of inertia of particles, $\omega_p$ is the angular velocity of lint particles' rotation, $T$ is the localized torque of particles, and $V_P$ is the particle velocity.

The lint pellet traction force is calculated by the formula [11]:

$$F_D = \frac{4}{3} C_D \frac{m_p \rho_p (v_g - v_p) |v_g - v_p|}{\rho_p d_p} \tag{6}$$

where $\rho_p$ is the density of particles, $C_D$ is the drag coefficient, and $d_p$ is the lint particle equivalent diameter.

The difference between particle gravity and buoyancy is:

$$F_{GB} = m_p (1 - \frac{\rho_g}{\rho_p}) g \tag{7}$$

The shear lift and rotational lift are perpendicular to the direction of the relative velocities of the particles and the gas phase; thus, the shear lift applied to the particles is calculated as:

$$F_{Sa} = 1.6515 d_p \mu_g C_{Sa} \text{Re}_s^{0.5} (|v_g - v_p| \times \omega_g) \tag{8}$$

$$\text{Re}_s = \frac{\rho_g d_p^2 |\omega_g|}{\mu_g} \tag{9}$$

where $\mu_g$ is the gas viscosity, $C_{Sa}$ is the shear lift coefficient, $\text{Re}_s$ is the shear Reynolds number for particles, and $\omega_g$ is the angular velocity of the gas' rotation.

The rotational lift realizes the relative motion between the particles and the gas phase, which can be expressed as:

$$F_{Ma} = \frac{\pi}{8} d_p^3 \rho_g (\omega_g - \omega_p) \times (v_g - v_p) \tag{10}$$

*3.2. CFD–DEM Coupling Parameter Analysis Settings*

3.2.1. Determination of Characteristic Parameters of The Cotton Pipeline

Pneumatic conveying throughout the whole process of cotton processing is the primary mode of transportation for seed cotton, lint, and other materials [27]. After the lint cleaning machine cleans cotton through the suction-type pneumatic conveying device, the cotton is sent to the baler in bales. The conveying device mainly comprises a lint suction nozzle, cotton collection branch pipe, cotton collection pipe, cotton collection machine, return air pipe, fan, dust collector, and other components and equipment, as shown in Figure 3.

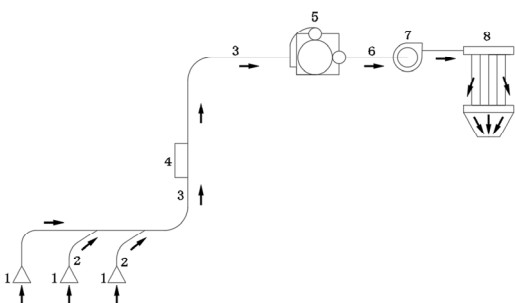

**Figure 3.** Cotton processing process flow chart: (1) lint nozzle, (2) cotton collection branch pipe, (3) cotton collection pipe, (4) online lint inspection device, (5) cotton collection machine, (6) return air pipe, (7) fans, (8) dust collector.

Cotton delivery in the processing plant mainly uses centrifugal suction fans, which are collected and transported through the negative-pressure cotton transport pipeline. In the lint cotton transportation chain, the length of the cotton pipeline is 15 m or less; cotton is transported in the pipeline at a speed of 12 m/s–15 m/s [28]; and the lint cotton transportation mixing ratio is 0.09~0.2. Based on a processing plant in Xinjiang, lint cotton conveying link field research was used to determine the centrifugal fan in the lint cotton conveying-related parameters as shown in Table 1.

**Table 1.** Reference configuration table for fans in lint transport.

| Fan Model | Fan Name | Number /Unit | Power of Motor /kW | Flow Rate /(m³/h) | Rotational Speed /(r/min) | Full Pressure (Pa) | Diameter of Air Inlet and Outlet (mm) |
|---|---|---|---|---|---|---|---|
| 4-72No.12C | Centrifugal fan | 1 | 45 | 49,641–69,481 | 1030 | 2318–1834 | 1000 |

### 3.2.2. Numerical Simulation Analysis of Unloaded Flow Field of Cotton Pipeline

The flow field simulation analysis focuses on the selection of the lint main pipe and the return air pipe. To determine the airflow velocity and negative-pressure distribution within the lint pneumatic conveying pipeline under no-load conditions, a numerical simulation analysis of the flow field was conducted. This analysis provides the necessary boundary parameters for the air–solid coupling in the CFD–DEM system. Figure 4 displays the three-dimensional structural dimensions of the cotton conveying pipeline.

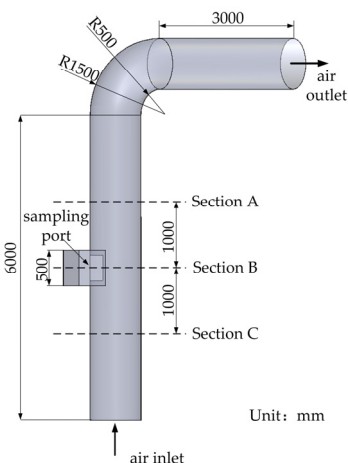

**Figure 4.** The three-dimensional structure of cotton conveying pipeline.

The mesh is the carrier of fluid simulation calculations, and the size and quantity of the mesh determine the accuracy and time of the solution. To verify the reasonableness of

the simulation model's mesh selection, this study meshed the cotton pipeline model with different densities; due to the large size of the model, a tetrahedral mesh was used to divide the model, and the mesh size was set by the proximity control function and the curvature control function; the mesh was fast using this mesh division, and at the same time, the model shape was regular, and the model could be reasonably divided by the edges and corners by these two control functions. The numbers of grids for the computational model were 1768, 3638, 6498, 8947, 52,822, 74,428, 104,737, 130,119, and 373,575. Figure 5 (the red stars are the data nodes; section B airflow velocity nodes correspond to different numbers of grids) represents the values of section B's airflow velocity, monitored as the model grid is refined while other conditions are kept the same, and it can be seen from the figure that the airflow velocity at section B changes very little when the number of grids exceeds 100,000, which means that continuing to increase the number of grids has less influence on the calculation results, and it can be considered that the 130,119 grids already meets the simulation requirements.

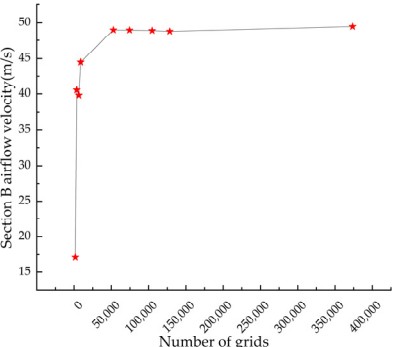

**Figure 5.** Grid independence test.

The cotton pipeline model was imported into Ansys Geometry and the fluid domain was delineated. Considering the shape and size of the model, the Proximity and Curvature function was used to mesh the model in Ansys Mesh, the Num Cells Across Gap was determined to be 8 layers, and the Curvature Normal Angle was determined to be 15°. Considering the boundary layer effect, the mesh near the pipe wall was encrypted in the thickness direction, Maximum Layers was set to 5 in Inflation, Growth Rate was set to 1.2, and Smooth Transition was selected in the Inflation Option to ensure a smooth transition of the mesh encryption; the mesh was divided as shown in Figure 6. As shown in Figure 6, the total number of meshes was 130,119, and the average value of Orthogonal Quality was 0.7136, which is more significant than 0.5, indicating that the quality of mesh division meets the requirements.

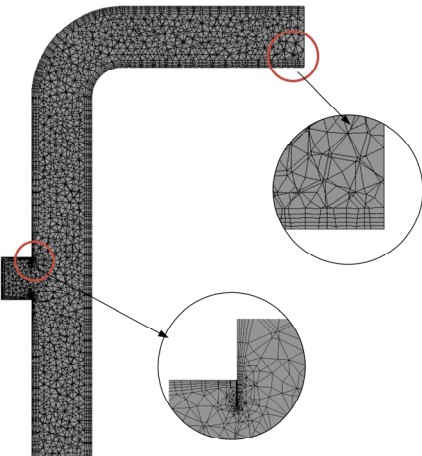

**Figure 6.** Schematic diagram of grid division.

The mesh model was imported into FLUENT fluid simulation software, and based on the characteristic parameters of the cotton pipeline, the external atmospheric pressure was set as the standard value, i.e., 101,325 Pa, the relative pressure at the aerodynamic inlet was set to be 0 Pa, and the pressure at the aerodynamic outlet was set to be $-2000$ Pa (the centrifugal blower provided negative pressure). Relying on the k–epsilon (2 eqn) turbulence model calculations to obtain the airflow in the pipe, the residuals of each index are below $10^{-4}$, and the calculation results converge. The simulation results are shown in Figures 7 and 8. After analysis, it can be seen that the average airflow velocity is 42.15 m/s and the average pressure is $-1297.71$ Pa when the pipeline is unloaded, and the maximum velocity is 45.2 m/s and the maximum pressure is $-1182.80$ Pa near the sampling port; the airflow velocity and pressure inside the pipeline tend to be uniformly distributed when the pipeline is unloaded, which contributes to the stable transportation of cotton lint. Meanwhile, the average velocity at the velocity inlet (section C) is 43.7 m/s, and the average pressure at the pressure outlet (section A) is $-1247.75$ Pa, which provides a theoretical basis for the parameter setting of the boundary conditions in the simulation of the cotton lint pneumatic conveying process by CFD–DEM.

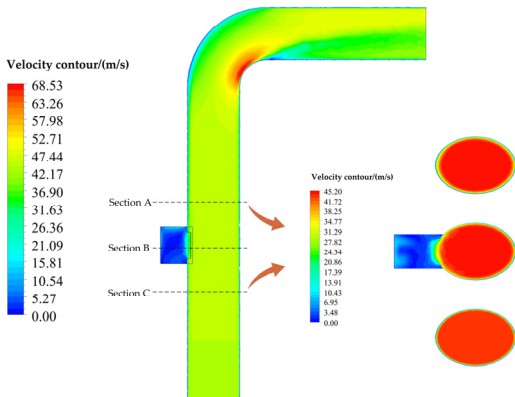

**Figure 7.** No-load flow field velocity cloud of cotton pipeline.

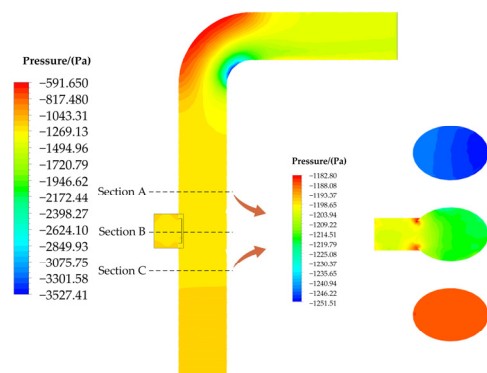

**Figure 8.** No-load flow field pressure cloud of cotton pipeline.

### 3.2.3. CFD–DEM Coupling Parameter Settings

The cotton pipeline with the presented sampling device (section A–section C) was intercepted for coupling analysis, and the parameters of the intercepting device are shown in Table 2.

**Table 2.** Quantitative sampling device-related parameters.

|  | **Parameters** | **Value** |
|---|---|---|
| Device simulation parameters | Pipe Diameter, mm | 1000 |
|  | Sampling port size, mm | $250 \times 250$ |
|  | Intercept length (section A–section B), mm | 2000 |
| Boundary conditions | Velocity inlet (cross section C), m/s | 43.7 |
|  | Pressure outlet (section A), Pa | −1247.75 |
|  | Pipeline pressure, Pa | −1297.71 |

In this EDEM–FLUENT coupled simulation test, we utilized FLUENT 2021 R1 and EDEM 2022 software to conduct the coupled simulation. The Eulerian–Lagrangian coupling model is chosen not only to realize the momentum exchange between the gas term and the lint particle term but also to take into account the effect of the particle volume on the gas continuum term, which can accurately analyze the interaction between the two gas–solid flows. The coupled flow process is shown in Figure 9. First, FLUENT iteratively calculates the time step of the airflow field. Suppose the current time step converges or reaches the preset number of iterative steps. In that case, the airflow force is applied to EDEM by compiling the coupling interface file, affecting the particle motion. Meanwhile, EDEM performs iterative calculations of the same time step by the discrete element method and imports the particle volume, position, and velocity information into FLUENT through the coupling interface, thereby calculating the interaction between the particles and the flow field. The entire coupling simulation process is achieved by iterating the calculation until the set convergence value is reached.

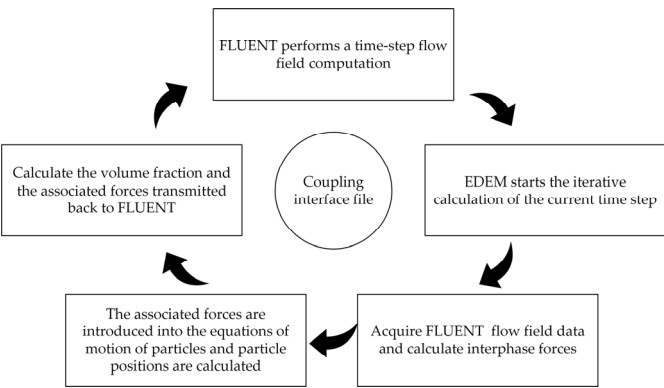

**Figure 9.** FLUENT–EDEM coupling calculation flow chart.

The grid file of the intercepted lint pipeline with the presented sampling device (section A–section C) was imported into EDEM 2022 software for solid-phase parameter setting, and the coupling interface file was used to start FLUENT software to realize the numerical simulation of the lint particle pneumatic conveying and sampling process. The force of the airflow field on the lint was simulated in FLUENT 2021 R1 software by checking the Freestream Equation traction model and Saffman Lift and Magnus Lift models in the coupling module. In EDEM, the solid-phase contact includes the contact between lint and lint and between lint and the wall; we chose the Hertz–Mindlin no slip, no sliding contact model reaction for the contact state between the particles and the wall.

In the CFD–DEM coupling process, lint particles are randomly generated in the velocity inlet, and the plant's generation rate and actual conditions are used to maintain consistency. Iteration occurs in EDEM 2022 software. The time step is $1 \times 10^{-6}$ s, and other parameters are shown in Table 3 [29–31].

**Table 3.** EDEM simulation parameters.

| | Parameters | Value/Formal |
|---|---|---|
| Material Properties | Poisson's ratio for lint | 0.4 |
| | Modulus of elasticity of lint/(Pa) | $2.4 \times 10^9$ |
| | Lint density/(kg·m$^{-3}$) | 400 |
| | Steel Poisson's ratio | 0.3 |
| | Steel shear modulus/(Pa) | $7 \times 10^{10}$ |
| | Steel density/(kg·m$^{-3}$) | 7850 |
| Exposure parameter | Lint–lint collision recovery factor | 0.05 |
| | Coefficient of static friction of lint–lint | 0.55 |
| | Lint–lint rolling friction coefficient | 0.15 |
| | Lint–steel collision recovery coefficient | 0.1 |
| | Lint–steel static friction coefficient | 0.45 |
| | Lint–steel rolling friction coefficient | 0.2 |
| Pellet Plants | Number of particles | limitless |
| | Particle generation rate (kg/s) | 1.69 |
| | Location and direction of particle generation | randomization |

The same mesh file was imported into FLUENT 2021 R1 software for flow simulation, and the turbulence model was selected as k-omega, which was solved by the Phase-Coupled SIMPLE algorithm. The time step of FLUENT was $1 \times 10^{-4}$ s (100 times of EDEM), the number of steps per coupling iteration was 30,000, and the total simulation time was 3 s; Max Iterations Time Step was 20, i.e., the maximum number of iterations per time step was 20; to extract the detailed information of particle movement as much as possible, the data were saved every 0.05 s in FLUENT. Max Iterations/Time Step was set to 20, i.e., each time step was iterated at most 20 times; to extract the motion information of the particles in as much detail as possible, the data were saved once every 0.05 s in FLUENT.

## 4. Simulation Results and Discussion

The analysis of the motion in a gas–solid multiphase flow system involves characterizing the motion of the fluid phase and the motion of the solid phase (particles). This study examines the impact of the cumulative weight of cotton samples on the quantitative sampling process of cotton lint in the processing chain. Specifically, it analyses the changes in the flow field, airflow pressure, airflow velocity, and motion state of cotton particles during the sampling process. In the CFD–DEM coupled simulation, the residuals of each index are below $10^{-3}$, and the calculation results converge, which can provide theoretical support for the design optimization of the quantitative sampling device.

### 4.1. Change in Pressure of Flow Field in Cotton Pipeline

The role of a negative-pressure centrifugal fan in the cotton pipeline involves the conveying of gas and lint particles. The interaction forces between these components are considered, and the flow field pressure is analyzed to evaluate the speed and stability of lint particle transport and the accumulation of particles in the sampling link. Additionally, the movement characteristics of the particles during the conveying process can be predicted. Therefore, an analysis of the flow field pressure at different moments during the re-sampling process of the cotton pipeline is conducted. Without sampling in the online detection device pipeline's flow field, the pressure remains stable at an average value of −1247.75 Pa. However, when the sampling plate is introduced into the pipeline, the pressure distribution along the pipeline is analyzed at specific time intervals of 0.05 s, 1 s, 2 s, and 3 s. These results are visually represented in Figure 10.

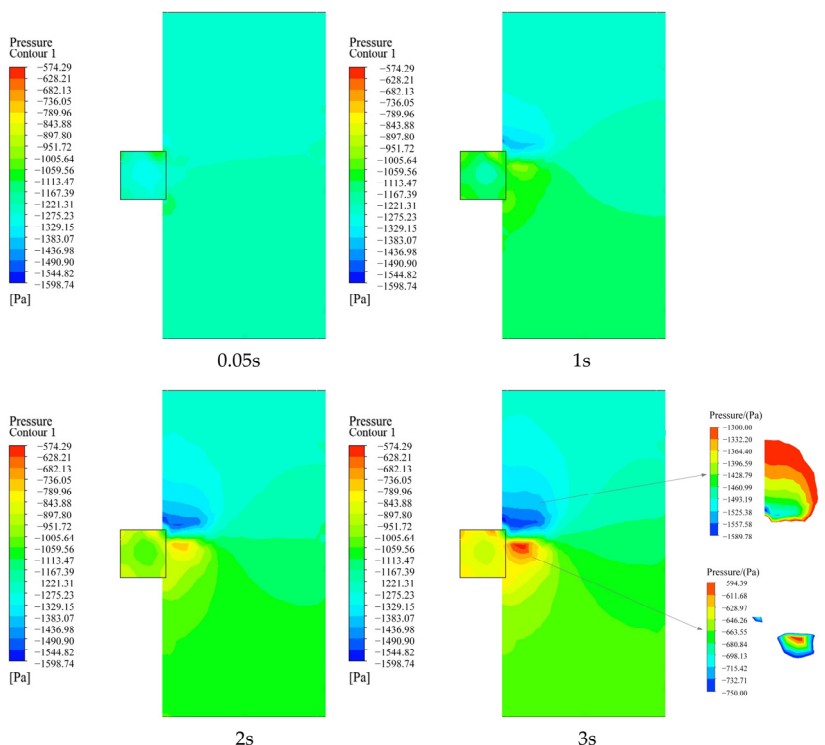

**Figure 10.** Flow field pressure clouds at different moments of the cotton pipeline.

The sample plate does not affect the initial pressure of the flow field at 0.05 s, and its pressure is around −1247.75 Pa, which is in a relatively stable state and helps to stabilize the transport and sampling of lint particles. After 0.05 s, the sampling plate affects the pressure of the flow field in the pipeline, and cotton particles on the sampling plate begin to pile up. The pressure at the upper end of the sample plate reduces as a result of the overhanging sampling plate's blocking impact on the airflow. In comparison, the pressure at the lower end increases. The low-pressure range at the higher end of the sample plate increases and then stabilizes as the sampling duration increases, while the high-pressure range at the lower end of the sampling plate increases and then stabilizes.

Through pressure map analysis conducted within a time frame of 1~3 s, it was observed that the sampling process resulted in a range of pressures. The upper end of the sampling plate exhibited the lowest pressure at −1598.74 Pa, while the lower end displayed the highest pressure at −574.29 Pa. The maximum pressure difference between the upper and lower ends of the sampling plate was found to be 1024.45 Pa. This pressure difference is crucial in facilitating the accumulation of cotton samples on the sampling plate, ensuring their stable arrangement. The sampling plate, made of galvanized steel and measuring 250 × 250 mm in size, was determined to be suitable for executing the negative-pressure conveying link sampling action. These findings provide theoretical support for the material design and selection of core sampling components used in online testing and inspection. The sample plate utilized in this study was constructed from a galvanized steel plate, measuring 250 × 250 mm. This specific size was determined to facilitate the successful execution of the negative-pressure conveying link sampling process. Additionally, the sample plate serves the purpose of offering theoretical justification for the design and selection of core sampling components utilized in online testing and inspection.

### 4.2. Velocity Change of Flow Field in Cotton Pipeline

The velocity of the flow field is a significant characteristic of pneumatic conveying systems. By examining the changes in and distribution of the flow field velocity, one can accurately evaluate the average flow rate of gas and lint particles within the pipeline and determine the maximum flow rate. This analysis is crucial in determining the stability of

the sampling device within the flow field and assessing its susceptibility to the impact of the fluent within the pipeline.

When the sampling plate is inserted into the online detection device without initiating sampling, the gas flow rate in the pipeline remains in a stable state, with an average steady flow rate of 43.7 m·s$^{-1}$. However, when the sampling plate is suspended in the pipeline and sampling is initiated, the gas flow rate distribution in the pipeline is analyzed at specific time intervals of 0.05 s, 1 s, 2 s, and 3 s. The corresponding results are depicted in Figure 11.

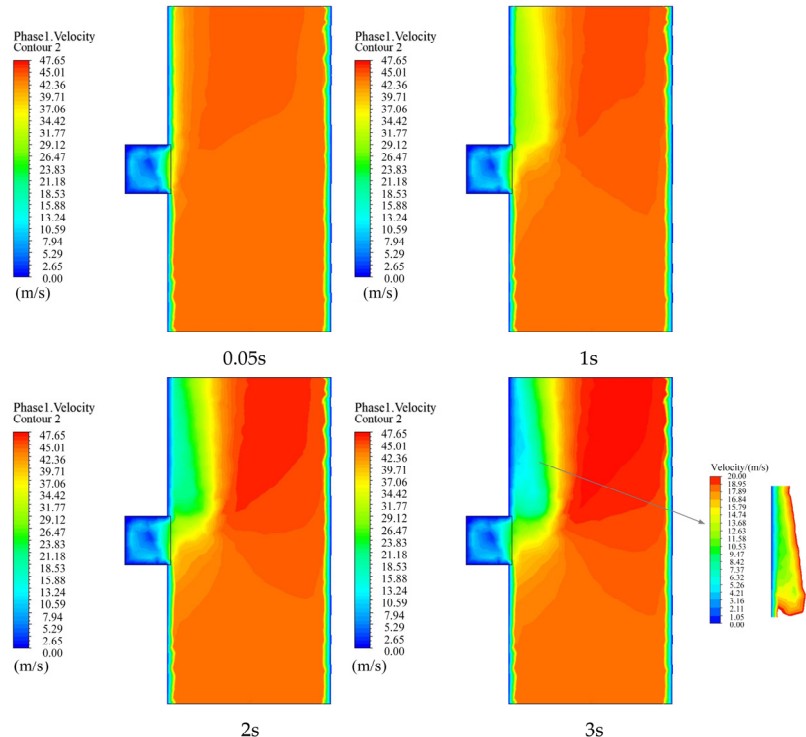

**Figure 11.** Flow velocity clouds of the flow field at different moments within the cotton pipeline.

When combined with the pressure map analysis, the sampling device does not affect the air velocity in the flow field at 0.05 s, and the average velocity is 43.7 m·s$^{-1}$. After 0.05 s, the sample plate creates a barrier effect in the pipeline, lowering the air velocity in the flow field near the sampling plate. The air velocity at the upper end of the sampling plate falls significantly with increasing sampling time, and the range of decline increases to the range of the sampling plate overhang length, according to the 1~3 s flow field velocity map analysis.

This study aims to investigate the variation in airflow rate over a 3 s interval. The low density and lightweight nature of cotton lint primarily influences this variation. The conveyance of the airflow rate within the pipeline's flow field is minimally affected by the presence of cotton lint. Additionally, the loose structure of cotton lint allows for the continuation of airflow through the sampling plate despite the accumulation of lint particles. Consequently, the lower end of the airflow assists in the sampling of cotton particles on the sampling plate.

### 4.3. Velocity Analysis of Lint Particles in Cotton Pipelines

The lint particles enter the negative-pressure cotton pipeline from the velocity intake with an initial velocity of 12 m/s in the linked simulation analysis. The cotton processing factory produces 27 bales of cotton lint each hour, and the total lint production is 6080 kg/h. Therefore, the simulated lint particle creation rate is 1.69 kg/s [32], consistent with actual production conditions. The movement of lint particles in the negative-pressure cotton pipeline under this condition was analyzed numerically to identify the sampling

process of lint particles as the movement speed changes and the impact on the sample plate's sampling.

The analysis involves sampling plates to observe the movement status of particles within a 3 s interval. The resulting diagram, depicted in Figure 12, displays the cloud of lint particle movement velocities at various time points. The analysis of the velocity of lint particles during the pneumatic conveying process with a duration of 0.05 s reveals that the particles experience a rapid increase in velocity to approximately 25 m/s when they enter the negative-pressure cotton pipeline. This indicates that pneumatic conveying is an effective method for quickly and efficiently transporting lint particles. Additionally, during the period of 1~3 s, the velocity of the lint particles remains stable at around 25 m/s and shows a tendency to stability. This observation, combined with an analysis of the air velocity in the negative-pressure conveying pipeline, suggests that the stable speed of the lint particles is approximately 59.31% of the velocity of the unloaded conveying wind. The analysis of the movement speed of lint particles within a 1~3 s timeframe reveals that these particles come into contact with a sampling plate once their speed decreases to 0 m/s. It is observed that the accumulation of particles on the sampling plate continues to increase, thereby demonstrating the feasibility of utilizing the sampling plate to collect cotton samples in a negative-pressure conveying environment. Moreover, this method ensures stable transportation of lint particles during the movement process, facilitating the effective capture of lint particles by the core of the sampling plate. The utilization of a sampling plate as the central component assists the sampling organization in achieving stable sample collection.

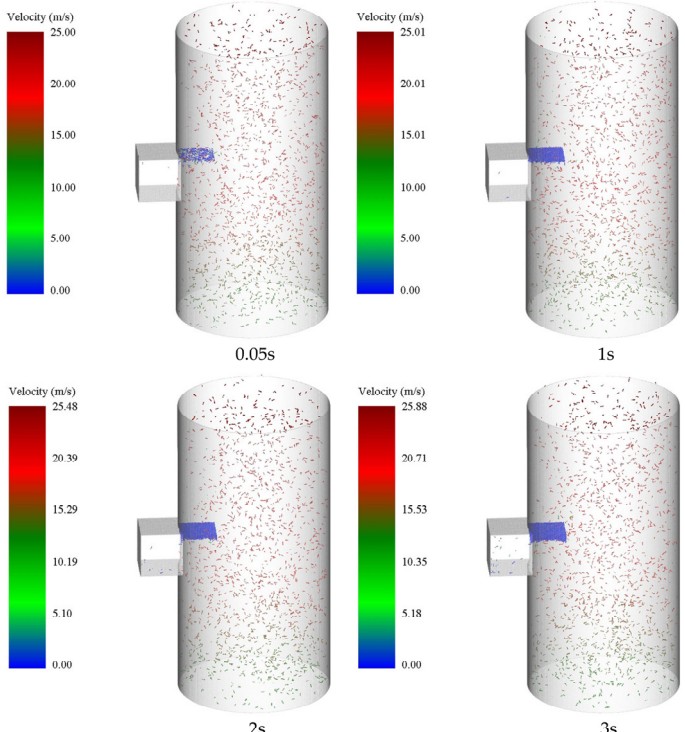

**Figure 12.** Cloud map of lint particle movement at different moments in the cotton pipeline.

### 4.4. Change in Weight of Lint Particles in the Sampling Device

The weight of the lint pile on the sample plate is a crucial evaluation metric in assessing the effectiveness of the sampling device in carrying out the quantitative sampling operation.

EDEM 2022 software employs the Grid Bin Group function to partition the sampling plate and collect cotton samples within a specified region. This facilitates the computation of the cumulative weight of lint particles in each divided area. Consequently, it becomes possible to ascertain the weight of lint particles on the sampling plate at various time

intervals during the collection process. Figure 13 illustrates the variations in lint weight collection within a 3 s timeframe.

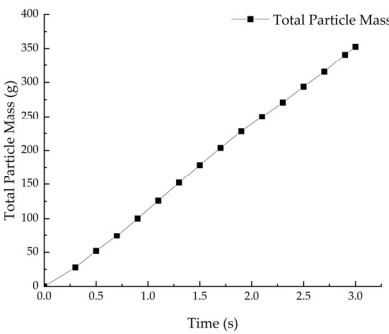

**Figure 13.** Cumulative change in lint weight in 3 s.

Analyzed from the above figure, the cotton lint's overall weight accumulation on the sample plate changes linearly; the transport of stable conditions of cotton lint particles amounts to an approximately 100 g/s collection speed for sampling. The results of the lint particle accumulation analysis at 1 s, 2 s, and 3 s are shown in Figure 14. At 1 s, the lint accumulation on the sampling plate is 108.739 g, at 1~2 s, it is 123.011 g, and at 2 s~3 s, it is 107.291 g. After analysis, under the condition of stable transportation of lint, the sampling device with the sampling plate as the core can realize quantitative sampling of lint in a certain period.

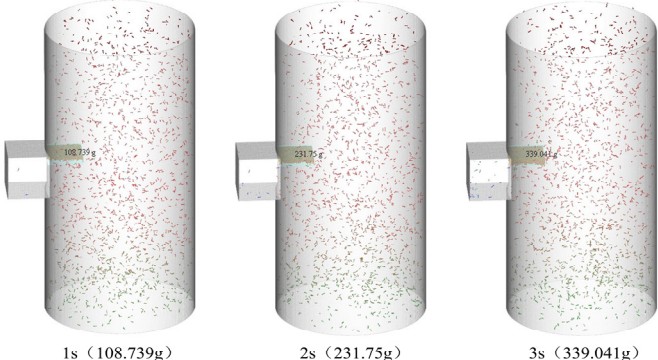

**Figure 14.** Weight of lint particle pile at different moments on the sampling plate.

## 5. Sample Machine Performance Test

To verify the stability and reliability of the quantitative sampling device and assess the sampling effect of the sampling device under the condition of uniform cotton flow, the quantitative sampling device was installed on a cotton pipeline to carry out a prototype performance test.

### 5.1. Pilot Program

5.1.1. Test Conditions

To assess the performance of our prototype, we installed a quantitative sampling device on a negative-pressure cotton pipeline at a processing plant in Xinjiang. This device allows us to transport lint through the pipeline during production, serving as the test object for our performance test. In field measurements, the cotton factory's pipeline suffers from poor tightness, resulting in resistance caused by factors such as pipeline resistance and local resistance (including inlet, lifting, elbow, tee, etc.). The actual unloaded wind speed in the cotton pipeline is measured to be 25~30 m/s, while the lint conveying speed is 8~12 m/s.

An MS6H-60CM30BZ3-20P4 servo motor (Xinjie Electric, Wuxi, China) is used in the quantitative sampling device. Selection of FAB060D-L2-14-50-70-M5-34 (TongLi Re-

ducer, Suzhou, China). We selected an L6D-C3-5kg-0.45B-FS-type load cell (AVIC Electro-Measurement). The size of the sampling plate was designed as a $250 \times 250$ mm square perforated galvanized steel plate, and the thickness of the sampling plate was 8 mm; other test components included a cotton collection box, nozzle, electric cabinet, touchscreen, and so on. The test setup is shown in Figure 15.

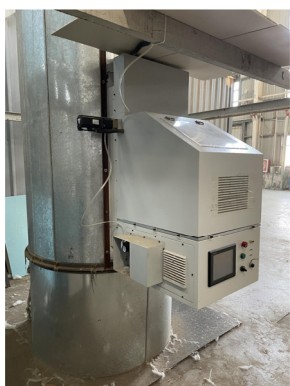

**Figure 15.** Quantitative sampling device.

### 5.1.2. Test Method

The quantitative sampling device is turned on, and the parts return to the starting position, as shown in Figure 16a. The device is now ready to sample with the adjustment plate for negative pressure positioned upwards and the sampling port open, as shown in Figure 16b. The sampling plate is then rotated 90° counterclockwise to enter the cotton pipeline, as shown in Figure 16c. Sampling begins, and the sampling plate remains in the pipeline with the cotton for 1 s. The lint is processed under negative pressure and accumulates below the sampling plate, indicating that the sampling is complete. This is shown in Figure 16d,e. After 1 s, the sampling plate rotates clockwise by 120°, allowing the cotton samples to enter the quantitative sampling device, as shown in Figure 16f. The cotton samples are then weighed. The negative-pressure regulator plate is closed to isolate the cotton pipeline environment from negative pressure. The cotton samples on the sampling plate are then transferred to the cotton collection box through the action of the pneumatic nozzle. The weight of the cotton samples is obtained using a load cell, as shown in Figure 16g.

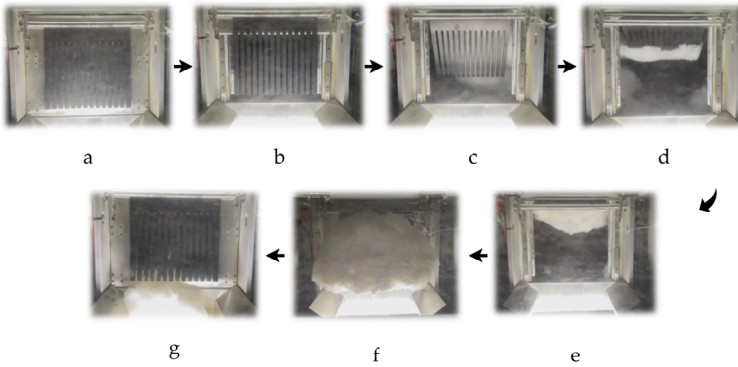

**Figure 16.** Test flow chart. (**a**) starting position, (**b**) sampling port open, (**c**) sample plate rotated 90° counterclockwise, (**d**) sample board stay, (**e**) cotton sample piling up, (**f**) sample plate rotated 120° clockwise, (**g**) getting cotton weight.

### 5.2. Results and Discussion

We repeated the sampling test 50 times and weighed the sample weight statistics; the results are shown in Figure 17. According to the national standard GB/T 6102.2-2012 [33]

raw cotton dampness test method, the sample quality is $50 \pm 5$ g. The test statistics show that the weight of the sample taken out is between 45~55 g. There are 43 groups, and the sampling qualification rate is 84%, which meets the sampling requirements.

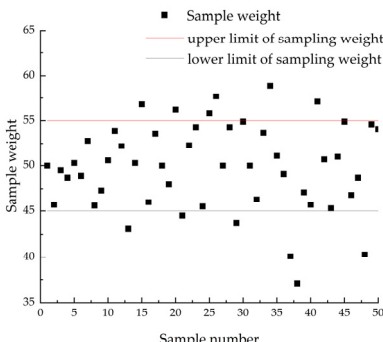

**Figure 17.** Sampling weight distribution chart.

## 6. Conclusions

1.  The prototype test shows that the sampling plate, as the core of the presented quantitative sampling device in cotton flow under uniform conditions, can achieve an 84% sampling pass rate during a specified period in a quantitative sampling study;
2.  During the conveying process in factories, there is a significant pressure difference of up to 1024.45 Pa when sampling plates move up and down. This pressure difference allows for the accumulation of cotton samples on the sampling plate, ensuring their stable placement. Additionally, this study provides theoretical support for the selection of the sampling plate's size and material, which are determined to be $250 \times 250$ mm and galvanized steel plate, respectively, to meet the requirements of actual sampling. This information is crucial for the design of core components in online testing equipment and material selection;
3.  Upon conducting an analysis, it was determined that the stable speed of cotton pipeline lint particles is 59.31% of the wind speed during unloaded conveying. This discrepancy in conveying wind speed within the conveying pipeline is attributed to variations in equipment parameters across cotton processing plants. Consequently, it is possible to calculate the lint particle conveying speed for different conveying wind speeds, thereby providing a theoretical foundation for determining the sampling time of the online testing device.

**Author Contributions:** Conceptualization, P.W. and H.W.; methodology, P.W.; experiments, P.W.; software, P.W.; formal analysis, P.W. and H.W.; investigation, P.W. and H.W.; validation, P.W., H.W., R.Z., B.H., R.H. and J.H.; writing—original draft preparation, P.W.; writing—review and editing, P.W. and H.W.; supervision, H.W. All authors have read and agreed to the published version of the manuscript.

**Funding:** This research was funded by the National Key R&D Plan of China (2022YFD2002400), the Science and Technology Bureau of Xinjiang Production and Construction Corps (2023AB014, 2022DB003), and the Science and Technology Planning Project of the 7th Division of Xinjiang Production and Construction Corps (QS2022005).

**Institutional Review Board Statement:** Not applicable.

**Data Availability Statement:** The data presented in this study are available on request from the corresponding author. The data are not publicly available due to privacy restrictions.

**Conflicts of Interest:** The authors declare no conflicts of interest.

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
