# Peer review of "Numerical Simulation of an Online Cotton Lint Sampling Device Using Coupled CFD–DEM Analysis"

_agriculture, doi:10.3390/agriculture14010127_

Round 1
Reviewer 1 Report (New Reviewer)
Comments and Suggestions for Authors
The scientific article "Numerical simulation of an in‐line cotton lint sampling device using coupled CFD‐DEM analysis" is devoted to cotton processing is the process of converting harvested seed cotton into lint by cleaning, ginning, and
cleaning the lint. The quantitative sampling procedure of gas‐solid two‐phase flow in a cotton pipeline was numerically simulated and experimentally analyzed by using Computational Fluid Dynamics (CFD) and Discrete Element Method (DEM).
Experimental and CFD results are in good agreement. The paper is well‐written and the quality of the figures is acceptable. The paper is interesting with some valuable conclusions and is recommended for possible publication in the Journal Agriculture. The experiments are carried out with a high degree of reliability and are of great scientific value. However, some revisions should be performed.
1. Explain the difference between the results obtained in this paper and the works of other authors. In the literature review, many works use one or another turbulence model. Critical analysis involves conversation with the topic of two
questions: what remains unsolved in the reference source and why this "something" is still unsolved (what are the reasons for this?). The authors believe "The analysis found that the stable transport velocity of cotton pipeline lint particles is 59.31% of the wind speed for unloaded transportation." Is this conclusion valid in all cases or are there any limitations? What are the limitations of this study? When changing what parameters will the velocity change? Can you compare this velocity with the experimental data?
2. There are several incorrect terms in the paper. The flow rate is not a velocity (line 220). There should be more information about "ω is the angular velocity of gas" and how it was calculated, or a reference to equation (10) by
other authors.
3. Better justify the selection of the turbulence model. Please provide more explanations of the reasons for the application of the "k‐w" or "k‐e" model. Row 315 ‐ k‐omega and row 265 ‐ k‐epsilon. Why did the authors change the turbulence model? The k‐epsilon model has shortcomings related to the simulation of the velocity in the boundary walls. The sampling weight distribution chart shows that the difference in weight between two samples can
be 2 times. How can you explain it?
In general, I think the manuscript can be accepted in its present form with minor changes.
Author Response
Please see the attachment.

Reviewer 2 Report (New Reviewer)
Comments and Suggestions for Authors
Title: Review Report on "Numerical Simulation of an In-line Cotton Lint Sampling Device Using Coupled CFD-DEM Analysis"
· The abstract of the research article provides a comprehensive overview of the study; however, it could benefit from being more concise. Streamlining the content will enhance its effectiveness in conveying the key objectives and outcomes of the research.
· In line 165, there is a minor typographical error; please correct the spelling of "conservation.
· The nomenclature section lacks the inclusion of ρg and ρp. Ensure their addition for clarity.
· In equation (4), it is noted that FDB should be corrected to FGB for accuracy.
· The reference for Equation 6 is missing. Ensure proper citation to support the equation's validity.
· A thorough check for correctness in Equation 9 is recommended to ensure accuracy and reliability of the presented results.
· Equation 8 requires a reference, and if it is empirical, its limitations should be discussed for a more comprehensive understanding.
· In Line 236, it is essential to specify the mesh type employed and provide justification for its selection. This information is crucial for readers to assess the reliability of the numerical simulations.
· The discretization techniques employed in the CFD simulation should be explicitly mentioned, along with the convergence criteria used. This information is critical for understanding the robustness of the numerical approach.
· It is imperative to state the convergence value achieved during the simulation. This detail adds transparency and enables the reader to assess the reliability of the numerical results.
· The selection of the k-omega turbulence model needs justification. Discuss the specific reasons behind choosing this model over others and its relevance to the study.
· Discussion of Results needs to be improved.
Comments on the Quality of English Language
NA
Author Response
Please see the attachment.

This manuscript is a resubmission of an earlier submission. The following is a list of the peer review reports and author responses from that submission.
Round 1
Reviewer 1 Report
Comments and Suggestions for Authors
The article submitted for review is interesting and concerns an important issue.
Some parts of the article require organization, and the number of literature sources analyzed and cited could be greater.
Below I present comments and questions that may be helpful in improving the manuscript:
- Please change the abstract. According to the guidelines of: The abstract should be a total of about 200 words maximum.
- Has there been any research published so far strictly on numerical simulations of the movement of lint particles in the flow field during pneumatic conveying? The authors point out that due to the shape and deformation, this is a difficult topic. Has it been undertaken so far?
- Please correct figure 1 - at this scale individual elements are invisible.
- In my opinion, a better idea is to show graphically (block diagram) the principle of operation.
- The dimensions in Figure 3 are illegible.
- Were calculations also performed for a different number of meshes?
- The description of Figures 5 and 6 is illegible (font too small)
Reviewer 2 Report
Comments and Suggestions for Authors
- The introduction section should be modified and more references should be provided in it.
- If experimental work and experimental testing have been done, it is suggested to add related descriptions and pictures in the article.
- The explanation provided about the numerical simulation is not enough. Please write this section more carefully.
- What was the solution method used in the numerical simulations?
Is the issue of mesh independence considered in the simulations? Be sure to add the corresponding diagram.
- How and with what tools have the authors confirmed the accuracy of the simulations?
- What was the purpose of the authors in presenting Figure 7? What information does it provide the reader?
- The presented results, especially from the numerical simulations, are very few and incomplete, and the authors must provide more results. Also, the results should go to graphs and tables instead of presenting contours.
Comments on the Quality of English Language
Moderate editing of English language required
Reviewer 3 Report
Comments and Suggestions for Authors
Lines 103-115: May be better to separate the sentence into several shorter complete sentences.
Line 110: “Realize” lower case. Same as line 129 “Vertical”. line 135 “Cotton”. lines 151, 163 “Where”
Lines 123-137: May be better to separate the sentence into several shorter complete sentences.
Lines 123-137: May create some illustrations/schematics to better describe the sampling process.
Make sure all the parameters are defined in the equations, like \rho_g, \rho_p
Table 3: Make sure the scientific notation is correctly expressed.
Table 3: How “Modulus of elasticity of lint=2.4e9” measured? It seems to be a pretty high value for cotton lint.
Table 3: “Exposure parameter” to “material interaction properties”.
Table 3: “Number of particles: limitless” Could you please provide some explanations for this?
Comments on the Quality of English Language
The manuscript would benefit from grammar editing. Especially pay attention to fragments are incomplete sentences.
Round 2
Reviewer 2 Report
Comments and Suggestions for Authors
The manuscript can be accepted in the present form.
Author Response
Thank you for your comments concerning our manuscript entitled “Numerical simulation and test of quantitative sampling process of lint cotton in processing link based on CFD-DEM” (No.: agriculture-2644134.).
Reviewer 3 Report
Comments and Suggestions for Authors
The content of the paper is good. However, the paper would benefit from language editing. Please double-check that all the sentences have verbs.
Comments on the Quality of English Language
The content of the paper is good. However, the paper would benefit from language editing. Please double-check that all the sentences have verbs.
Author Response
Thank you for your comments concerning our manuscript entitled “Numerical simulation and test of quantitative sampling process of lint cotton in processing link based on CFD-DEM” (No.: agriculture-2644134.). We have added missing verbs to sentences and completed editing of English language. The revised version has been uploaded.